



# The Global Streamflow Indices and Metadata Archive (GSIM) – Part 2: Quality Control, Time-series Indices and Homogeneity Assessment

Lukas Gudmundsson[1], Hong X. Do[1], Michael Leonard[2], Seth Westra[2], Sonia I. Seneviratne[1]

[1]Institute for Atmospheric and Climate Science, Department of Environmental Systems Science, ETH Zurich,
Universitaetstrasse 16, Zurich 8092, Switzerland
[2]School of Civil, Environmental and Mining Engineering, University of Adelaide, Adelaide, Australia

*Correspondence to*: Lukas Gudmundsson (lukas.gudmundsson@env.ethz.ch)

**Abstract.**

This is Part 2 of a two-paper series presenting the Global Streamflow Indices and Metadata Archive (GSIM), which is a collection of daily streamflow observations at more than 30000 stations around the world. While Part 1 describes the data collection process as well as the generation of auxiliary catchment data (e.g. catchment boundary, land-cover, mean climate), Part 2 introduces a set of quality controlled time series indices representing (i) the water balance, (ii) the seasonal cycle, (iii) low-flows and (iv) floods. To this end we first consider the quality of individual daily records using a combination of quality flags of the data providers and automated screening methods. Subsequently streamflow time series indices are computed for yearly, seasonal and monthly resolution. The paper closes with a generalized assessment of the homogeneity of all generated streamflow time series indices, which can be used to select time series that are suitable for a specific task. The presented global set of streamflow time series indices is made freely available at https://iacweb.ethz.ch/staff/lukasgu/GSIM/GSIM_indices.zip and is expected to foster global freshwater research, by acting as a ground-truth for model validation or as a basis for assessing the role of human impacts on the terrestrial water cycle.

## 1 Introduction

Although terrestrial freshwater variables are essential components of the Earth system and a prerequisite for societal development, the availability of relevant in-situ observations at the global scale has been limited. Until now, most relevant in-situ observations have been held by national and regional authorities, and despite best efforts, international data centres only have access to a small subset of the full observed record (Do et al., submitted). This situation stands in contrast to the fact that monitoring data are increasingly being made publicly available through regional and national authorities (Do et al., submitted). In this paper series, we present an international collection of river and streamflow observations that covers more than 30000 stations around the globe, highlighting the fact that these are among the best monitored variables of the terrestrial water cycle (Fekete et al., 2012; Fekete et al., 2015; Gudmundsson and Seneviratne, 2015; Hannah et al., 2011). Part 1 of the paper series (Do et al., submitted) documents the data-collection process together with a meta-data base that allows users to recreate the collection from the original data sources. In addition, Part 1 of this paper series also presents auxiliary data including catchment





boundaries delineated from global digital elevation models as well as selected properties (e.g. land-cover, climate) of these catchments.

While the data collection outlined in Part 1 (Do et al., submitted) increases the spatial and temporal availability of streamflow records at the global scale, it is important to also consider the quality of the data. This is especially relevant for this merged

data-product which combines information from several data bases, which might have been set up with different objectives. In addition, changes in instrumentation as well as human impacts such as stream straightening or flow regulations can have pronounced effects on the observed record. Establishing a database of quality controlled streamflow observations is therefore essential for many applications, including e.g. the need to evaluate the increasing number of continental and global scale models that have emerged in recent decades (Beck et al., 2017; Gudmundsson et al., 2012a; Gudmundsson et al., 2012b;

Haddeland et al., 2011; Zaitchik et al., 2010) and the assessment of human impacts on the terrestrial water cycle (Alkama et al., 2013; Barnett et al., 2008; Destouni et al., 2013; Hegerl et al., 2015; Hidalgo et al., 2009; Jaramillo and Destouni, 2015). While there have been significant efforts in the climatological community to share and standardise transnational weather observations as well derivative data products (Becker et al., 2013; Dee et al., 2011; Harris et al., 2014; Haylock et al., 2008; Poli et al., 2016), the hydrological community has traditionally been reticent to adopt regional or global approaches, instead

focusing predominantly on the catchment scale. A more concerted and coordinated effort to understand the quality of streamflow observations across the globe provides significant opportunities for improving large scale hydrological models that support understanding of global water budgets. This paper initiates the process of evaluating, analysing and documenting the quality of observed streamflow time-series, providing a method for increasing the reliability and ongoing value of the database. For doing so, this paper expands on previous research (Gudmundsson and Seneviratne, 2016) and applies a set of transparent

and reproducible methods to evaluate the quality of the considered records.

One limitation of the newly assembled collection of daily river and streamflow time series is that publication of unprocessed daily values is restricted for some of the original data sources. To nevertheless be able to publish relevant information on observational streamflow, we therefore present here processed data in the form of time-series indices that capture essential aspects of (i) the water balance, (ii) seasonality, (iii) low-flows and (iv) floods. The approach of publishing time-series indices

instead of raw daily values, is adapted from the CCl/WCRP/JCOMM Expert Team on Climate Change Detection and Indices (ETCCDI) (https://www.wcrp-climate.org/data-etccdi), who have developed this approach to make relevant climate information publicly available in cases were access to raw daily values is restricted. The ETCCDI has focussed on indices characterizing changes in extreme precipitation and temperature, based on a core collection of indices proposed by Frich et al. (2002). Both Klein Tank et al. (2009) and Zhang et al. (2011) provide additional background on the usage and computation of

the ETCCDI indices. Klein Tank et al. (2009) also provides guidelines for quality control of the raw daily input data, index computation and assessment of time-series homogeneity.

The use of time series indices for characterizing the temporal evolution of selected river flow characteristics is also common practice in the hydrological literature. Typically used time series indices include mean annual flows (e.g. Kumar et al., 2009; Milly et al., 2005; Small et al., 2006; Stahl et al., 2010; Stahl et al., 2012), indices that can be used to characterize changes in





the seasonal cycle (e.g. Blöschl et al., 2017; Cunderlik and Ouarda, 2009; Ehsanzadeh and Adamowski, 2010; Hidalgo et al., 2009; Moore et al., 2007; Rauscher et al., 2008; Regonda et al., 2005; Stewart et al., 2005), time series of annual percentiles (e.g. Gudmundsson et al., 2011; Lins and Slack, 1999; Zhang et al., 2001), flood indices (e.g. Blöschl et al., 2017; Hodgkins et al., 2017; Kumar et al., 2009; Kundzewicz et al., 2005; Lins and Slack, 1999; McCabe and Wolock, 2002; Small et al., 2006;

Svensson et al., 2005; Zhang et al., 2001) and low-flow indicators (e.g. Hisdal et al., 2001; Lins and Slack, 1999; McCabe and Wolock, 2002; Small et al., 2006; Stahl et al., 2010; Stahl et al., 2012; Svensson et al., 2005; Tallaksen et al., 1997; Zhang et al., 2001).

In addition, several studies have focussed on collections of hydrological signatures (or flow characteristics) that are designed to summarize long-term properties of observed river and streamflow (e.g. 2013; Beck et al., 2015; Olden and Poff, 2003;

Sawicz et al., 2011; Sawicz et al., 2014; Westerberg et al., 2016). These hydrological signatures include e.g. mean annual flow, flow percentiles, characteristics of the flow duration curves, indications of seasonality and the base flow index. These signatures are typically derived from all daily values in a long time window (e.g. the base flow index computed from all daily values between 1985 to 2010). This is an important structural difference if compared to time-series indices, which are typically computed every year, every season or every month (e.g. time series of annual maxima).

The following sections build upon this previous research and present a collection of quality controlled river and streamflow time series indices. For doing so, we first introduce an approach to check the quality of individual daily observations using a combination of information provided with the original data and data-driven procedures. Subsequently we present a collection of time-series indices that can be computed for yearly, seasonal and monthly resolution. The paper closes with an assessment of the statistical homogeneity of the newly derived indices that allows users to filter the published data according to pre-defined

eligibility criteria.

## 2 Quality control (QC) of daily values

### 2.1 Strategy for QC of daily values

As the considered data stem from several sources, some of which have a complex history, it is difficult to a-priori judge the quality of individual records. Ideally, each of the considered series would be accompanied with detailed information on (a) the

station properties (e.g., information on sensors or the design of the gauging weir) and (b) on the credibility of individual daily values. However, this information is often not available or difficult to access and only some of the original data sources provide daily quality flags (Table 1). In addition, the large number of languages involved and the sheer quantity of gauging stations renders a detailed manual assessment infeasible. Nevertheless, it is essential to appraise the quality of individual observations prior to any assessment. As some of the considered time series come with daily quality flags, while some others do not, the

two cases are treated independently.





## 2.2 Quality control of daily values if reliable flags are provided

As noted in Do et al. (submitted), some of considered databases provide quality control (QC) flags for daily values that provide a distinction between reliable and suspect observations (Table 1). To allow for a combined assessment, the original QC flags were translated into unified values that separate *suspect* from *reliable* values (Table 2). This step is necessary for consistency
since some databases provide a variety of QC flags to indicate suspect cases but neither the same flags nor the level of fidelity is available across all databases. Regarding the Global Runoff Data Centre (GRDC), while QC flags are available in the EWA and GRDC files, the GRDC advises not to use them. In these cases, the time series are treated as if no QC flags were provided. Some data bases do not provide QC flags for every time step (Table 2), in these cases time steps without original QC flags were assumed to be *reliable* as long as at least one time step was flagged in the respective series.

## 2.3 Quality control of daily values if no reliable flags are available

For original time series files for which no QC flags are available or for which there is advice against using available QC flags by the data providers (GRDC and EWA), automated techniques can be used to classify the reliability of individual daily data points using simple and reproducible tests. The following three criteria are based on a previously used procedure (Gudmundsson and Seneviratne, 2016) and were developed on the basis of techniques described in Reek et al. (1992) and ECA
& D Project Team and Royal Netherlands Meteorological Institute (2013; later referred to as EAC&D13) and further refined using suggestions on outlier detection for index calculation by Klein Tank et al. (2009):

1. Days for which $Q < 0$ are flagged as *suspect*, where $Q$ denotes a daily streamflow value. The rationale underlying this rule is that streamflow values smaller than zero are non-physical (Gudmundsson and Seneviratne, 2016).

2. Daily values with more than 10 consecutive equal values larger than zero are flagged as *suspect*. This rule is motivated
by the fact that many days with consecutive streamflow values often occur in case of instrument failure (e.g. damaged sensor, ice-jams) or flow regulations. The threshold of 10 days is a compromise chosen to account for the possibility that consecutive equal observations reflect the true values e.g. if day-to-day fluctuations are below the sensitivity of the employed sensor (Gudmundsson and Seneviratne, 2016).

3. Following a previously suggested approach for evaluating temperature series (Klein Tank et al., 2009), daily
streamflow values are declared as outliers if values of *log(Q+0.01)* are larger or smaller than the mean value of *log(Q+0.01)* plus or minus four times the standard deviation of *log(Q+0.01)* computed for that calendar day for the entire length of the series. The mean and standard deviation are computed for a 5-day window centred on the calendar day to ensure that a sufficient amount of data is considered. The log-transformation is used to account for the skewness of the distribution of daily streamflow values and 0.01 was added because the logarithm of zero is undefined. Outliers
are flagged as *suspect*. The rationale underlying this rule is that unusually large or small values are often associated with observational issues. The four standard-deviation threshold is a compromise, aiming at screening out outliers that could come from instrument malfunction, while not flagging extreme floods or low flows.



An example of the outcome of this automated quality control of daily observations is shown in Figure 1, which displays daily streamflow observations and highlights time-steps that did not pass the three abovementioned criteria. Note that the outlier detection did not screen out extreme floods, but values that were unusually large for the respective time of the yeaer.

## 3 Streamflow indices

### 3.1 General considerations, design rules and reliability

#### 3.1.1 General considerations

In the following, we present a set of streamflow time-series indices that are designed to facilitate the analysis of (i) changes in the regional water balance, (ii) changes in the seasonal cycle, (iii) floods and (iv) low flows. Many of the indices presented below were previously used in the scientific literature and wherever possible a selection of relevant references will be included. Note also that the selection was limited to indices that can be computed without a base period, which excludes many; examples include "the number of days in a year, or season, for which daily values exceed a time-of-year-dependent threshold" (Zhang et al., 2005), drought deficit volumes (Loon and Anne, 2015; Tallaksen et al., 1997) and anomalies with respect to a climatological normal (McKee et al.; Shukla and Wood, 2008). There are two reasons for excluding these indices: First, regional differences in temporal coverage hinder an unambiguous identification of a common base period that can be used around the globe. Second, it is now well established that indices that depend on a base period are prone to inhomogeneities if the base period is shorter than the considered series (Sippel et al., 2015; Zhang et al., 2005). Although both analytical (Sippel et al., 2015) and non-parametric (Zhang et al., 2005) solutions exist to mitigate this problem, we chose not to include indices that require a base period. This is because the available solutions either depend on strong normality assumptions (Sippel et al., 2015) or are computationally intensive (Zhang et al., 2005), which implies that the time series indices cannot be easily extended when new data become available.

#### 3.1.2 Design rules for index calculation

The design rules for calculating time-series indices closely follow the recommendations of ECA&D13. Before index calculation, all daily values that are flagged as *suspect* by the daily QC procedure are set to missing and indices are computed using the remaining data points. All indices are computed on yearly time-steps, while some indices are also computed with seasonal and monthly resolution. Seasons are defined as December-January-February (DJF), March-April-May (MAM), June-July-August (JJA) and September-October-November (SON). The reason for not computing all indices for seasonal and monthly resolution is related to (i) the fact that some indices are only interpretable on annual time scales or (ii) to issues related to the amount of data required for reliable computation.





### 3.1.3 Reliability of index values

Not all daily time steps have observations and some daily observations have been flagged as *suspect* and were therefore removed. Consequently yearly, seasonal and monthly index values are not equally reliable. To allow users to judge upon the reliability of index values at individual time-steps, the number of daily values used for index calculation at each time step are

provided. Following the recommendations of ECA&D13, the following rules on daily data availability can be applied to identify *reliable* index values:

1. Index values at a **yearly** time step are reliable if at least 350 daily observations are declared *reliable*.
2. Index values at a **seasonal** time step are reliable if at least 85 daily observations are declared *reliable*.
3. Index values at a **yearly** time step are reliable if at least 25 daily observations are declared *reliable*.

## 3.2 Description of indices

### 3.2.1 Mean daily streamflow

*Abbreviation:* MEAN

*Units:* [$m^3 sec^{-1}$]

*Time resolution:* yearly, seasonal, monthly.

*Definition:* Arithmetic mean of daily streamflow.

*Comments:* Mean daily streamflow is a commonly used water-balance measure and often used as a proxy for renewable freshwater resources (Oki and Kanae, 2006; Shiklomanov et al., 2004; Vörösmarty et al., 2000). Observed timeseries of mean yearly or monthly streamflow has e.g. been subject to trend analysis at regional to continental scales (e.g. Kumar et al., 2009; Lettenmaier et al., 1994; Lins and Slack, 1999; Milly et al., 2005; Small et al., 2006; Stahl et al., 2010; Stahl et al., 2012).

### 3.2.2 Standard deviation of daily streamflow

*Abbreviation:* SD

*Units:* [$m^3 sec^{-1}$]

*Time resolution:* yearly, seasonal, monthly.

*Definition:* Standard deviation of daily streamflow.

*Comments:* The standard deviation of daily streamflow provides information on the total variability for each yearly, seasonal and monthly time step. This index therefore includes information related to floods and low flows as well as the amplitude of the annual cycle (yearly only). We are not aware of any study analysing time series of the standard-deviation of daily streamflow.

### 3.2.3 Coefficient of variation of daily streamflow

*Abbreviation:* CV

*Units:* [-]

*Time resolution:* yearly, seasonal, monthly.

*Definition:* Standard deviation of daily streamflow divided by the mean daily streamflow (SD/MEAN).

*Comments:* The coefficient of variation of daily streamflow is a relative measure of daily variability. In contrast to SD, CV is
independent of the mean flow and does hence allow for an isolated assessment of day-to-day streamflow variability. We are
not aware of any study analysing time series of the coefficient of variation of daily streamflow.

### 3.2.4 Interquartile range of daily streamflow

*Abbreviation:* IQR

*Units:* [$m^3$ $sec^{-1}$]

*Time resolution:* yearly, seasonal, monthly.

*Definition:* $75^{th}$ – $25^{th}$ percentile of daily streamflow.

*Comment:* The inter quartile range is a measure of day-to-day streamflow variability. Through its definition as the difference
between the $75^{th}$ and the $25^{th}$ percentile, the IQR provides information on the width of the centre of the distribution and is less
sensitive to extreme outliers than SD or CV. We are not aware of any study analysing time series of the standard-deviation of
daily streamflow.

### 3.2.5 Minimum daily streamflow

*Abbreviation:* MIN

*Units:* [$m^3$ $sec^{-1}$]

*Time resolution:* yearly, seasonal, monthly.

*Definition:* Minimum value of daily streamflow.

*Comment:* Minimum daily streamflow is a regularly used low-flow indicator. Especially the yearly minimum has been used
widely as it is an easy to interpret measure and lends itself to analysis in the framework of the generalized extreme value
distribution (Tallaksen and van Lanen, 2004). Annual minimum streamflow series are also commonly subject to large-scale
trend analysis (e.g. Kumar et al., 2009; Lins and Slack, 1999; McCabe and Wolock, 2002; Zhang et al., 2001).

### 3.2.6 Maximum daily streamflow

*Abbreviation:* MAX

*Units:* [$m^3$ $sec^{-1}$]

*Time resolution:* yearly, seasonal, monthly.

*Definition:* Maximum value of daily streamflow.

*Comments:* Maximum daily streamflow is a widely used indicator for high-flows and floods. Especially annual maximum
time series are regularly considered as they allow for a straight forward interpretation and can easily be analysed through the



generalized extreme value distribution (Katz et al., 2002). Time series of annual maximum streamflow have been subject to regional and global trend assessments (e.g. Do et al., 2017; Hall et al., 2015; Kumar et al., 2009; Kundzewicz et al., 2005; Lins and Slack, 1999; McCabe and Wolock, 2002; Small et al., 2006; Zhang et al., 2001).

### 3.2.7 Minimum 7-day mean streamflow

*Abbreviation:* MIN7

*Units:* [m$^3$ sec$^{-1}$]

*Time resolution:* yearly, seasonal, monthly.

*Definition:* Minimum 7-day arithmetic mean streamflow. For computation, the complete daily time series are first smoothed with a backward looking moving average with a seven-day window. Subsequently, the minimum value for each yearly, seasonal or monthly period is determined.

*Comments:* Time series of minimum 7-day mean streamflow have been repeatedly used as a low-flow and drought metric. Through the smoothing operation, MIN7 is less sensitive to small day-to-day fluctuations, but focusses on sustained periods with limited water availability. MIN7 time series have e.g. been subject to large scale trend assessments (Kumar et al., 2009; Small et al., 2006; Stahl et al., 2010; Svensson et al., 2005).

### 3.2.8 Maximum 7-day mean streamflow

*Abbreviation:* MAX7

*Units:* [m$^3$ sec$^{-1}$]

*Time resolution:* yearly, seasonal, monthly.

*Definition:* Maximum 7-day arithmetic mean streamflow. For computation, the complete daily time series are first smoothed with a backward looking moving average with a seven-day window. Subsequently, the maximum value for each yearly, seasonal or monthly period is determined.

*Comments:* Time series of 7-day mean maximum streamflow do not focus on the highest water levels ever recorded, but rather on sustained periods of very high flow. Time series of MAX7 have e.g. been used to assess streamflow trends in India (Kumar et al., 2009; Stahl et al., 2012).

### 3.2.9 10[th], 20[th], 30[th], 40[th], 50[th], 60[th], 70[th], 80[th] and 90[th] percentile of daily streamflow

*Abbreviation:* P10, P20, P30, P40, P50, P60, P70, P80, P90

*Units:* [m$^3$ sec$^{-1}$]

*Time resolution:* yearly, seasonal.

*Definition:* Percentile values of daily streamflow computed for each yearly and seasonal period, where low percentiles (e.g. 10[th] percentile) correspond to low-flows.

*Comments:* Percentiles of daily streamflow provide together with MIN and MAX an approximation of the empirical cumulative distribution function (ECDF) of daily streamflow for each considered seasonal or yearly time period. These indices are not provided on monthly resolution, as it appears to be excessive to compute percentiles in 10% steps based on 28 to 31 daily values. Note also, that an alternative definition of the ECDF is also referred to as the flow-duration curve (FDC) in the hydrological literature. The difference between the ECDF and the FDC is that the FDC uses an inverse definition of percentiles (exceedance frequencies), such that high values correspond to low-flows (Tallaksen and van Lanen, 2004; Vogel and Fennessey, 1994). Besides approximations of the ECDF, the percentile series can be used to characterize "moderate extremes" (Zhang et al., 2011), i.e. very high or very low values that can occur several times each year and are hence more robust to quantify. Sets of annual percentile series have for example been used to investigate regional low- and high-flow dynamics in Europe (Gudmundsson et al., 2011) and have been subject to regional scale trend assessments (Lins and Slack, 1999; McCabe and Wolock, 2002; Zhang et al., 2001).

### 3.2.10 Centre Timing

*Abbreviation:* CT

*Units:* [doy]

*Time resolution:* yearly.

*Definition:* The day of the year (doy) at which 50% of the annual flow is reached. The index is computed for calendar years, where 1 denotes 1st January.

*Comments:* The centre timing is an index that is sensitive to changes in the seasonal cycle. Lower values indicate that more than half of the annual discharge has occurred earlier in the year. That means, that values smaller or equal than 182 would correspond to a year for with at least half of the streamflow volume has occurred in the first half of the year. Note that CT is usually defined for hydrological years in the literature and that the precise definition of CT can vary between studies (Hidalgo et al., 2009; Moore et al., 2007; Rauscher et al., 2008; Regonda et al., 2005; Stewart et al., 2005). Here we compute CT for calendar years to ensure consistency with the remaining indices and because the definition of the hydrological year depends on local climate conditions. Time series of CT have been used to assess changes in the timing of the seasonal cycle of streamflow in several regional studies (Hidalgo et al., 2009; Moore et al., 2007; Rauscher et al., 2008; Regonda et al., 2005; Stewart et al., 2005).

### 3.2.11 Day of minimum streamflow

*Abbreviation:* DOYMIN

*Units:* [doy]

*Time resolution:* yearly.

*Definition:* The day of the year (doy) at which the minimum flow occurred, where 1 denotes 1st January. The maximum value is 365 for normal years and 366 for leap-years.

*Comments:* The timing of annual minimum flow can provide valuable information on the processes underlying low-flows. For example, in snowy regions, the minimum flow often occurs in the winter months, whereas in other regions minimum flows occur in the season with low precipitation and large atmospheric water demand. We are not aware of any study that is explicitly analysing time series of DOYMIN.

### 3.2.12 Day of maximum streamflow

*Abbreviation:* DOYMAX

*Units:* [doy]

*Time resolution:* yearly.

*Definition:* The day of the year (doy) at which the maximum flow occurred, where 1 denotes 1[st] January. The maximum value is 365 for normal years and 366 for leap-years.

*Comments:* The timing of annual maximum streamflow can be a valuable indicator for the flood generating processes. In cold regions annual, maximum flow is often associated with snow melt, while in other regions it may be associated with intense convective precipitation during the warm season or soil moisture. Time series of DOYMAX have for example been used to assess trends in the timing of floods in Europe (Blöschl et al., 2017) and Canada (Cunderlik and Ouarda, 2009).

### 3.2.13 Day of minimum 7-day mean streamflow

*Abbreviation:* DOYMIN7

*Units:* [doy]

*Time resolution:* yearly.

*Definition:* Day of the year (doy) at which the minimum 7-day arithmetic mean streamflow occurred, where 1 denotes 1[st] January. The maximum value is 365 for normal years and 366 for leap-years. For computation, the daily time series is first smoothed using a backward looking moving average with a 7-day window length. Subsequently, the day of the minimum of each year is determined.

*Comments:* Overall the interpretation of DOYMIN7 is analogue to the interpretation of DOYMIN. Note, however, that DOXMIN7 is representative for a seven-day period of sustained low-flows and is less sensitive to outliers. We are not aware of any study that is explicitly analysing time series of DOYMIN.

### 3.2.14 Day of maximum 7-day mean streamflow

*Abbreviation:* DOYMAX7

*Units:* [doy]

*Time resolution:* yearly.

*Definition:* Day of the year (doy) at which the maximum 7-day arithmetic mean streamflow occurred, where 1 denotes 1[st] January. The maximum value is 365 for normal years and 366 for leap-years. For computation, the daily time series is first





smoothed using a backward looking moving average with a 7-day window length. Subsequently, the Julian day of the maximum of each year is determined.

*Comments:* Generally, the interpretation of DOYMAX7 is analogue to the interpretation of DOYMAX, although DOYMAX7 represents a one-week period of sustained high flows and is less sensitive to outliers. We are not aware of any study that is

explicitly analysing time series of DOYMAX7.

### 3.2.15 Gini coefficient

*Abbreviation:* GINI

*Units:* [-]

*Time resolution:* yearly.

*Definition:* For daily runoff values $q$ of each year, that are sorted with index $i$ in increasing order such that $q_i \leq q_{i+1}$ GINI is

defined as $\frac{1}{n}\left( n + 1 - 2\left( \frac{\sum_{i=1}^{n}(n+1-1)q_i}{\sum_{i=1}^{n} q_i} \right) \right)$, where $n$ is the number data points available for that year. The Gini coefficient ranges

from 0 to 1.  Values of 0 indicate uniform distribution of flows throughout the time period (i.e. year), whereas values close to 1 indicate that all the flows over a reference period occurs on a single day.

*Comments:* The Gini coefficient is a metric that was originally established in economic sciences as a measure for economic

inequality (Ceriani and Verme, 2012). It is a measure of dispersion that is not dependent on the absolute value of the variable under consideration and can be interpreted as a measure of the variability implied by the flow duration curve. It is therefore, like the CV a relative variability measure that can easily be compared among different regions. Although we are not aware of any study investigating annual GINI time series derived from streamflow, relevant applications to observed precipitation (Rajah et al., 2014) and global hydrological model output (Masaki et al., 2014) are emerging.

**3.3 Example time series**

To provide a first impression of the considered indices, Figure 2 shows all indices at annual resolution for Wiese at Zell, located in south western Germany. In addition, Figure 3 shows the daily observations of the same river together with the MEAN at monthly, seasonal and yearly resolutions.

**3.4 Temporal coverage of yearly, seasonal and annual indices**

The above mentioned daily quality control (section 2) as well as the ECA&D13 criteria for judging upon the reliability of yearly, seasonal or monthly index values (section 3.1.3) imply that the space-time coverage of the index data is not equal to the coverage of the original daily time series. Figure 4 shows both the number of yearly, seasonal and monthly time steps that are available at each location together with the fraction of time steps that were classified as reliable. Overall Figure 4 highlights that the collection of the longest time series can be found in the United States, central and northern Europe, Russia and southern

Africa. The fraction of reliable yearly, seasonal and monthly time steps is larges in the United States and Europe. Furthermore,



it should be noted that the fraction of reliable time steps is lowest for yearly indices. This is related to the fact that full years are deemed unreliable when less than 350 valid observations are used for computation (following the ECA&D13 rules). Note however, that the relatively strict ECA&D13 rules can be relaxed depending on user needs.

## 4 Homogeneity assessment

### 4.1 Methods for homogeneity assessment

#### 4.1.1 Homogeneity tests

Any environmental time series can be subject to inhomogeneities, i.e. unnatural sudden shifts in their statistical moments. In the simplest case, such inhomogeneities could be a jump in the mean between two time periods (see Figure 5, top), but also changes in variability (e.g. reduced peak-flows) or shifts in higher-order moments can be issues. The reasons for such inhomogeneities in streamflow time series are manifold, but they can "be related to changes in instrumentation, gauge restoration, recalibration of rating curves, flow regulation or channel engineering" (Gudmundsson and Seneviratne, 2016). As all the above-mentioned factors can be detrimental to scientific investigation, it is essential to check time series against inhomogeneities. Here we apply a previously utilised collection of tests (Gudmundsson and Seneviratne, 2016), which is recommended by ECA&D13 and has been thoroughly tested for temperature and precipitation indices (Wijngaard et al., 2003). This collection of tests contains (i) the standard normal homogeneity test (Alexandersson, 1986), (ii) the Buishand range test (Buishand, 1982), (iii) the Pettitt test (Pettitt, 1979), and (iv) the von Neumann ratio test (von Neumann, 1941). For the application of the abovementioned collection of tests, we rely on tables that provide critical values of the test-statistics for a given sample size that have been determined using Monte Carlo methods (ECA&D13). These tables only report critical values for a sample size of 20 and larger. Therefore, the tests can only be applied if at least 20 yearly, monthly or seasonal time-steps are available. Prior to homogeneity testing, yearly, seasonal and monthly index values that are classified as unreliable according to ECA&D13 (see section 3.1.3) are set to missing. Missing values were removed after pre-whitening of yearly, seasonal and monthly index time-series (see section 4.1.2).

#### 4.1.2 Pre-whitening

As the considered homogeneity tests rely at least on the assumption that the data are stationary and independent and identically distributed, all indices are pre-processed (pre-whitened) aiming at reducing effects of (i) trends, (ii) seasonality, and (iii) serial correlation. For the pre-whitening procedure, linear trends and mean seasonal cycles were removed using a linear least-squares regression model which captures both the trend and the mean values as $x = b + a \times t$, where $b$ is the intercept, $a$ is the trend and $t$ is time:

1. For **yearly** indices, the linear model is fitted to and subtracted from the complete time series. This results in a time series with zero mean and no linear trend.





2. For **seasonal** indices, the linear model is fitted to and subtracted from the time series for each season (DJF, MAM, JJA, SON) individually. This results in a time series with seasonal resolution in which each season has a zero mean and no linear trend.

3. For **monthly** indices, the linear model is fitted to and extracted from the time series for each month (January, February, …) individually. This results in time series with monthly resolution in which each month has a zero mean and no linear trend.

As the detrended and de-seasonalised time series may still exhibit serial correlation, they were further pre-whitened by fitting a lag-1 autoregressive model and then obtaining the residuals, which are then subjected to the homogeneity analysis (Burn and Elnur, 2002; Chu et al., 2013; Gudmundsson and Seneviratne, 2016). The lag-1 autoregressive model is fitted using maximum likelihood estimation.

### 4.1.3 Classification of station homogeneity

To effectively combine the information of the four considered homogeneity tests, we classify the homogeneity of yearly, monthly and seasonal time-series indices following recommendations of ECA&D13:

1. *useful:* 1 or 0 tests reject the null hypothesis at the 1% level.
2. *doubtful*: 2 tests reject the null hypothesis at the 1% level.
3. *suspect*: 3 or 4 tests reject the null hypothesis at the 1% level.

In addition, we also introduce the following categories to account for special circumstances that can occur in this large-scale application:

4. *not sufficient data:* less than 20 yearly, seasonal or monthly reliable index values are available.
5. *constant:* all yearly, seasonal or monthly time-steps have the same value.
6. *error:* an error (e.g.: numerical convergence issues) occurred at any processing step.

### 4.2 Homogeneity testing of all yearly, seasonal and monthly time-series indices

The homogeneity analysis is applied to all indices at yearly, seasonal and monthly resolution. The rationale for applying the four tests to all indices individually is that inhomogeneities at a particular location might be relevant only for a sub-set of indices, while other indices are not affected. For example, it is possible that a change in instrumentation will affect peak flows, while low flows are not affected. For this homogeneity assessment, all yearly, seasonal and monthly time-steps that are classified as reliable (section 3.1.3) are considered. This results in a conservative assessment as (i) strict data-availability criteria are applied and (ii) because inhomogeneities could occur in a time window not relevant for a study. Therefore, the presented results can be used for a general overview on time-series homogeneity but its suitability of should always be re-considered prior to specific applications.

Figure 5 illustrates the results of the homogeneity assessment for the MEAN index at the North Umpqua River in the US. The top panel shows the monthly MEAN index, which displays a sudden jump after the first third of the record. This jump may

e.g. be the result of upstream flow regulation and would be detrimental for climatological investigations. The lower panel shows the time-series after the above-mentioned pre-whitening procedure was applied. The seasonal cycle is effectively removed and obtaining the residuals from the lag-1 autoregressive model did reduce the magnitude of the sudden jump. Note also, the spurious trend, which is a consequence of applying de-trending in case of strong sudden shifts in the mean.

Nevertheless, three of the four considered tests identify this inhomogeneity at the 0.01 significance level and the series is classified as *suspect*.

Global summaries of the number of stations in different homogeneity classes are shown in Figure 6 and Figure 7. Owing to the reduced number of time steps, the homogeneity testing could only be applied for approximately half of the locations. Nevertheless, the homogeneity assessment highlights the other half of the yearly indices can be considered "useful" at many

locations. Only a small number of the low-flow indices (e.g. MIN, P10, P20, P30) had "constant" values and other issues were rarely detected. For both seasonal and monthly resolution, the number of stations with sufficient data for homogeneity assessment increased significantly, although it is important to recall that the homogeneity tests were in many cases applied to relatively short records (i.e. at least 20 seasons or 20 months respectively). Most of the seasonal and monthly time series with sufficient data are classified as "useful" but also a number of "doubtful" and "suspect" values were detected. At a few locations,

low-flow indices had constant values.

## 5 Data availability

The data described in this paper are available as a compressed zip-archive containing (i) a readme file, (ii) all time-series indices and (iii) the results of all homogeneity tests. For the peer-reviewing phase the data can be freely downloaded at https://iacweb.ethz.ch/staff/lukasgu/GSIM/GSIM_indices.zip. Conditional on acceptance of this article, the data will be made

available on the www.pangea.de data server and will be accessible through a digital object identifier (doi).

## 6 Summary and conclusions

Together with Do et al. (submitted) (Part 1), this paper presents the The Global Streamflow Indices and Metadata Archive (GSIM), which is a unique collection of streamflow observations at more than 30000 stations around the globe. In Part 1 (Do et al., submitted) of the paper series we focused on the collection and merging of freely available streamflow data worldwide.

Part 1 also introduced shape-files of catchment boundaries together with essential catchment properties such as land cover, topography and mean climatic conditions. As not all data-providers allow for a free distribution of unprocessed daily values, we followed in Part 2 an approach that has been established through the ETCCDI in climate research (Klein Tank et al., 2009; Zhang et al., 2011) and introduced a set of time-series indices that can be used to assess the water balance, seasonality, low-flows and floods.





To ensure the reliability of the published data, we first evaluated the quality of individual daily values through a combination of quality flags developed by the data providers and a transparent numerical screening approach. Subsequently streamflow time series indices were computed for yearly, seasonal and monthly resolution. In the last step, we assessed the homogeneity of all the indices using reproducible methods, aiming at aiding potential users to gauge the suitability of individual time series

for their research endeavours. To serve the international scientific community, this unprecedented data set, containing streamflow time series indices at more than 30000 gauging stations around the world, is made freely available.

There is a strong imperative for ongoing activities that not only preserve and mine the historical instrumental record, but add value to it. These varied activities might include: providing new analyses that improve the quality and understanding of the existing database; developing new automated methods that can be used systematically to maintain or improve the quality of

the instrumental record; providing additional streamflow observations from missing or currently inaccessible datasets; lobbying to have closed databases be made open and accessible; deriving new products though better ground-truthing of remote-sensed variables or reanalysis from hydrological models; and arranging for coordinated systems to improve updating, storage and documenting of existing data.

There are numerous unsettled scientific questions at the global scale that this dataset has the potential to support. For example,

there are unresolved questions around the relationship between trends in rainfall extremes and hydrological extremes (Do et al., 2017; Westra et al., 2013), as well as developing a better understanding of the influence of human activities on the hydrological cycle more broadly (Barnett et al., 2008; Blöschl et al., 2017; Destouni et al., 2013; Hegerl et al., 2015; Jaramillo and Destouni, 2015). Expanding upon recent methodological developments (Gudmundsson and Seneviratne, 2016, 2015), the newly assembled data may act as a basis for developing gridded global-scale observations based data products. There are also

likely to be many applications in fields as diverse as hydro-ecology, water quality modelling, environmental assessment and socio-hydrology. We therefore expect the presented data to be a valuable source of information to answer pending questions in global freshwater research, e.g in the context of the World Climate Research Program Grand Challenge on Water Availability (Trenberth and Asrar, 2014) or the international research efforts on "Change in hydrology and society" (Montanari et al., 2013).

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



**Tables**

**Table 1: Quality flags of daily values of all data bases that enter the G-SIM collection (see Do et al. (submitted)).**

| Database | Quality code |
|---|---|
| GRDC | Not recommended by data provider. There are four flags: |
| | -999 - missing data, no correction |
| | 1 - corrected data, no method specified |
| | 99 - usage not recommended by the provider |
| | 900 - calculated from daily water level |
| EWA | Not useful (similar to GRDC) |
| ARCTICNET | Qc flag not provided |
| GAME | Qc flag not provided |
| CHDP | Qc flag not provided |
| USGS | Four categories: |
| | A: value has been validated to be published |
| | A:e: value was estimated and validated to be published |
| | P and P:e: Provisional data |
| BOM | Flags were provided for each data point. |
| | There are five categories documented: |
| | A (flag 10): best available data |
| | B (flag 90): compromised to represent the parameter |
| | C (flag 110): estimated value |
| | E (flag 140): quality is not known |
| | F (flag 210): poor quality or missing |
| | Flag '-1' also presents to indicate missing value |
| HYDAT | Five categories: |
| | A: Partial Day (numeric value 1) |
| | B: Ice Conditions (numeric value 2) |
| | D: Dry (numeric value 3) |
| | E: Estimated (numeric value 4) |
| | S: Sample(s) collected this day (numeric value 5) |
| WRIS | Qc flag not provided |
| ANA | Flags were described in Portuguese. Below are my interpretation with the help of Google Translate |
| | 0: No description |
| | 1: Real measurement |
| | 2: Estimated |
| | 3: Doubtful |
| | 4: Dry |
| MLIT | Qc flag not provided |
| AFD | Qc flag not provided |



**Table 2: Translation of daily quality control (qc) flags of the original data bases (Table 1) to standardized values prior to the calculation of indices. Note that the Global Runoff Data Centre advises not to consider the qc flags in the GRDC and EWA files. Note also that some data basis (HYDAT, ANA) do not provide qc flags for all daily data.**

| Dataset | qc flag not provided | use not recommended by data provider | reliable | suspect |
|---|---|---|---|---|
| GRDC | - | All data points | - | - |
| EWA | - | All data points | - | - |
| ARCTICNET | All data points | - | - | - |
| GAME | All data points | - | - | - |
| CHDP | All data points | - | - | - |
| USGS | - | - | 'A' and 'A:e' (approved data) | 'P' and 'P:e' data (provisional data) |
| BOM | - | - | A (table below) | B, C, D, F (table below) |
| HYDAT | Other data points | - | B, D, S | A, E |
| WRIS | All data points | - | - | - |
| ANA | 0, no value | - | 1, 4 | 2, 3 |
| MLIT | All data points | - | - | - |
| AFD | All data points | - | - | - |



**Figures**

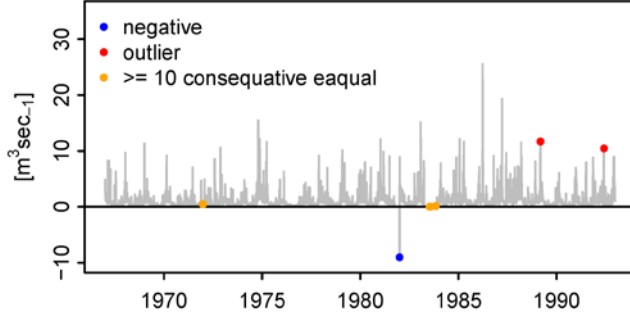

**Figure 1: Time series of Zwalm at Nederzwalm in Belgium. Time-steps that did not pass the three formal daily quality control criteria are highlighted. See text for details**



**Figure 2: All considered indices at yearly resolution, shown for river Wiese at Zell, south western Germany. Yearly values are only displayed if they contain at least 350 reliable daily observations. See text for details on units, interpretation and reliability classification.**



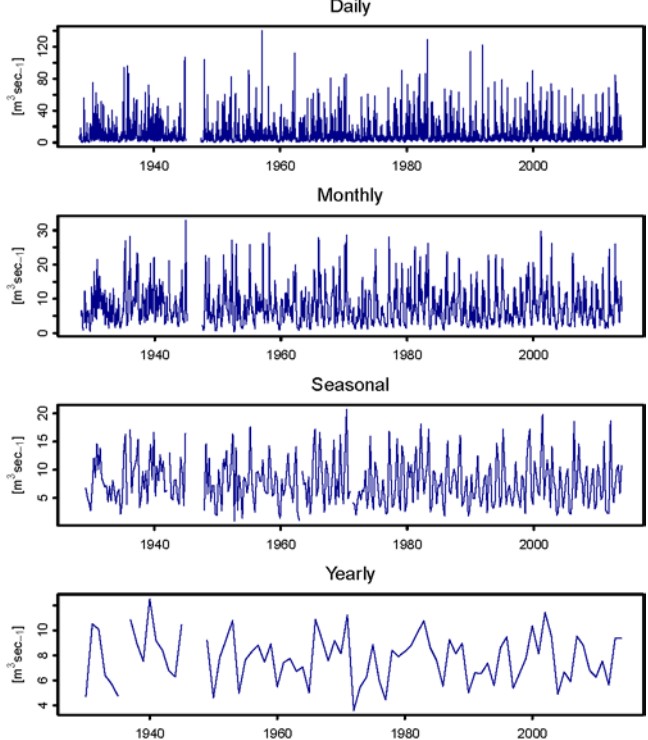

Figure 3: Daily values as well as monthly, seasonal, and yearly MEAN for river Wiese at Zell, south western Germany. Daily values are only shown if they are deemed reliable in the daily quality control procedure. Index values are only displayed if they fulfil the ECA&D13 data availability criteria. See text for details.



**Figure 4: Temporal coverage of indices at yearly, seasonal and monthly resolution. Left: Number of time steps, ranging from the first to the last non-missing index value. Right: Fraction of yearly, seasonal and monthly time steps that are classified as reliable using the ECA&D13 data availability criteria.**



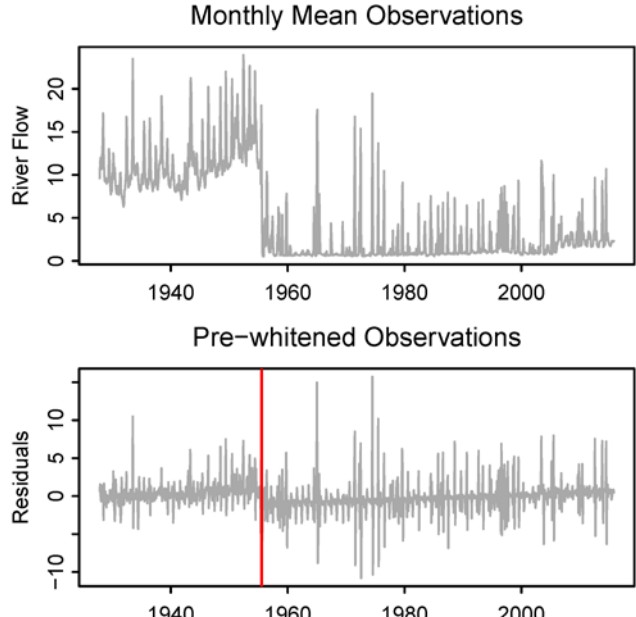

**Figure 5: Homogenity assessment of monthly mean flow of North Umpqua River, US. Top panel: Monthly mean observations. Bottom panel: Pre-whitened observations together with the time-step at which the standard normal homogeneity test, the Buishand range test and the Pettitt test identified a breakpoint at the 0.01 significance level.**



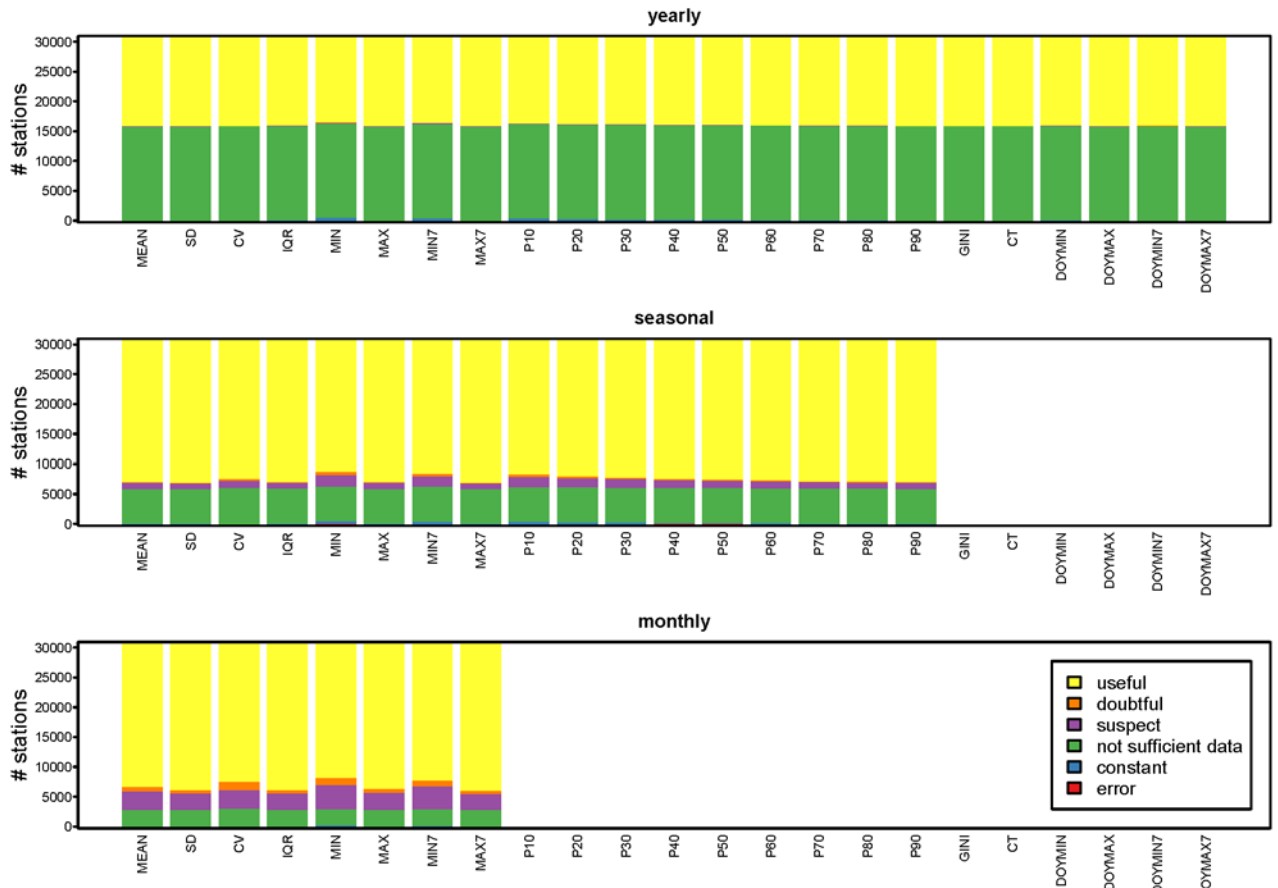

**Figure 6: Global summary of the homogeneity analysis for all considered indices at yearly, seasonal and monthly resolution. Shown are the number of stations that are classified as (1) useful, (2) doubtful, (3) suspect, (4) not sufficient data, (5) constant and (6) error according to section 4.1.3. Note that all 6 categories do occur, although some of them are rare and thus barely visible in the figure.**





**Figure 7: Spatial overview on index homogeneity at yearly, seasonal and monthly resolution. Left: stations at which all indices were classified as "useful". Right: stations that have sufficient data for homogeneity assessment for each time resolution.**