# Peer review of "The Global Streamflow Indices and Metadata Archive (GSIM) – Part 2: Quality Control, Time-series Indices and Homogeneity Assessment"

_Earth System Science Data, 2017_

## Referee Comment (RC1) · W. Grabs (Referee) · 18 Dec 2017

ESSD- 2017- 104

Review

Wolfgang E. Grabs

10 be consistent with number of station in first paper (35.000 stations)

25 Although monitoring data are increasingly posted on national and regional websites,

this does not relate to entire time series of data. In many cases, data are presented in graphical format only to prevent potential misuse

p 2, para 5 The data contained in different data bases also stem from different sources and often unknown quality control procedures

2, 25 typ-o: "In cases where access..."

3, 15 A critical question is whether this data set is a closed data set or whether there is a strategy to keep the data set alive through updates In the latter case, a possible procedure could be outlined how to apply the calculation of the indices to updated time series. Possibly, a tool kit could be proposed to calculate indices on new or updated datasets

4,5 In its latest version (summer 2017), GRDC is no longer providing flags and continues to abstain from flagging data

4,10 Routine QC of data by GRDC (and most other data centres) is limited to plausibility checks and correction of obvious errors

4, 10-15 It should be noted that base or reference periods are an important feature to characterize deviations from base period statistics that is vital for a number of management decisions in water management. The use of climate normal is such an example and efforts are underway to link climate normal to hydrological normals calculated for rivers. Although such normal are not globally valid, they are regionally significant

4,15 The notion that "time series indices cannot be extended when new data becomes available" is a serious constraint that limits the utility of the data set and the approach chosen. Moreover, a closed data set is aging fast. This needs to be discussed more in the paper. There is also the danger that such a data set becomes "orphaned" and continues to be used over a long period of time even when better data set have become available, similar like the RivDis Data of UNESCO more than 20 years ago that still is being used although it contains errors and is long-since superseded
9,20 typ-o: "...to a year for which...."

Quality control

The indices should be divided in those that actually express data quality and those that are significant for science applications such as water balance indications etc. A transparent approach should be taken to categorize data as a result of the data quality indices, making however clear that QC procedures cannot replace the responsibility of data owners and providers who in the first place provided data to data centres. The QC indices alone are already a valuable asset of the data set provided, guiding researchers and practitioners in the selection of data sets for their purposes.

Discuss minimum length of time series needed for specific applications. For climate studies, i.e. WMO recommends a minimum length of 30 years.

The paper should discuss the robustness of the chosen indices on time series with differing qualities. A procedure could be proposed or recommended to check robustness of indices under varying conditions

―――――――――――――――――

---

## Referee Comment (RC2) · Anonymous Referee #2 · 31 Dec 2017

General comments

I found this second, and (acknowledged) more difficult-to-do, part useful and publishable in general. But it reads less finished than the first part of the companion papers and my assessment is it still requires a bit more work on text and structure to be as influential in the end as perhaps can be on the community's use of the presented dataset. Again, of course the data quality control approach chosen, selection of indices for the dataset and the homogeneity tests chosen are subjective and pragmatic and not everyone needs to agree with it and it will never be possible to satisfy everyone's needs,

so I don't see a point in criticising individual choices and decisions made here. What I think, however, would help the paper's acceptance and impact in particular regarding the use of the quality assessments given is to bring out the assessment of the result of these better, incl. for example a nicer clearer presentation and assessment of the summary statistics of quality flags, homogeneity tests and indices (Figs 4-7 are a bit uninspiring and very simple - maybe combining maps and box plots somehow or something like that) and more comment/assessment on the patterns on the maps that will allow at least some guidance to the data usage. Partly this is can perhaps be done by just a bit of a better organisation and presentation of the material that is there, but also by a more structured discussion, also linking to metadata from part 1.

It wasn't always clear to me, which information I will find in the dataset and which are only steps of production described here. This could be make clearer overall and in some cases removing repetitions may help (first an overview paragraph, then reading the same thing again later in the individual steps is unnecessary). Section 6 in particular is a bit random in what is covered and highlighted and thus presents not a strong conclusion. I suggest to give this some thought and better organise and bring out the highlights. This is perhaps an editorial decision, but I find sections 3.2.1 to 3.2.15 not a correct nor a very useful text format as they contain neither a list nor paragraphed text and the use of the many subheadings is an unnecessary waste of space. Tables have been invented to reduce repetitive headings/descriptors. So why not a table with the name, abbreviation, unit, resolution and definition followed then by the more descriptive text paragraphs providing additional info. Alternatively, just a series of paragraphs always starting in similarly structured sentences would do as well. The selection of example studies is a bit random. Is it really necessary? This paper is supposed to describe the dataset created and it may be enough to use some of these references exemplary in a summary-motivation for the selection of indices or rather in a discussion on possible use of the dataset information.

Generally, the manuscript will also require another careful proofread to correct several typos and some inconsistent formatting (italic or quotes for dataset flags/categories/...confusing!), some terminology (examples below), tenses (e.g. what 'was' done to the data - use part tense consistently - and what 'is' provided in the dataset - use present tense consistently) and notation (examples below - not an exhaustive list). In particular: see Journal's Manuscript guidelines for symbols, exponents and units (e.g change sec to s and make format exponents as superscripts (most figures))

Selected specific comments

3.3.1. Isn't 'reference period' the more commonly used terminology (instead of base period)?

3.1.3 Requirements for number of valid data to estimate a 'reliable index'. These are very subjective, which I know is a necessary pragmatic solution. However, it creates a bias to less 'reliable' indices in climates streamflow gauging isn't possible or meaningful part of the year (seasonally dry climates and cold climates). This needs to be discussed.

Harmonize the current mix of Q/C, Qc, qc, and define what is meant by it initially as common definitions vary.

27 typo: appraise

Fig. 1 Typo in legend: 'equal', change axis label sec –> s and proper superscript (also in Fig. 3). I suggest to zoom in more as like this there is actually little to see.

Fig. 3 Since a) daily data won't be provided by the dataset anyway if I understand correctly and b) one cannot see anything in the daily graph anyway, I suggest to remove it from Fig.3

Similar to part 1, but perhaps even more so here, are the global maps. At that size and resolution it's impossible to see anything and not enough credit is given to why these difference may not simply reflect a lack of data but an inherent feature of the variable

covered and which may not be present or measurable (see earlier comment). When I zoom in I see grids rather than station location points, but didn't read anything on gridding the point information. This is not acceptable and needs to be changed or very clearly described.

---

## Author Comment (AC1) · 26 Jan 2018

**Response to Reviewer #1 (W. Grabs)**

We would like to thank W. Grabs for the open comments, which will clearly help to improve the presented manuscript. In the following we provide point by point replies to the reviewer's suggestions. For the sake of clarity, we first repeat the reviewer's comments in blue before including our response.

10 be consistent with number of station in first paper (35.000 stations)

Thanks for noting, we will ensure consistency in the revised manuscript

25 Although monitoring data are increasingly posted on national and regional websites, this does not relate to entire time series of data. In many cases, data are presented in graphical format only to prevent potential misuse

Thank you for noting that not all regional/national streamflow archive are available for download. Nevertheless, we think that it is noteworthy to emphasise that a substantial number of time series is now publicly accessible. This is documented in Part 1 of this paper series.

p 2, para 5 The data contained in different data bases also stem from different sources and often unknown quality control procedures

Thank you for noting, we will highlight this issue in the revised manuscript

2, 25 typ-o: "In cases where access. . ."

Thank you for noting, the paper will be carefully checked for typos before re-submission.

3, 15 A critical question is whether this data set is a closed data set or whether there is a strategy to keep the data set alive through updates In the latter case, a possible procedure could be outlined how to apply the calculation of the indices to updated time series. Possibly, a tool kit could be proposed to calculate indices on new or updated datasets

As mentioned in our response to the review for Part 1, GSIM will likely become a closed data set. The reason for this is that we do not have the resources for regular updates. However, we would welcome any initiative aiming at keeping the presented collection alive.

4,5 In its latest version (summer 2017), GRDC is no longer providing flags and continues to abstain from flagging data

Thank you for this valuable information. We will incorporate this in the revised manuscript.

4,10 Routine QC of data by GRDC (and most other data centres) is limited to plausibility checks and correction of obvious errors

Thank you for this information. We will mention this in the revised manuscript. We will also put this into context of our daily QC criteria which are also focussing on plausibility checks.

4, 10-15 It should be noted that base or reference periods are an important feature to characterize deviations from base period statistics that is vital for a number of management decisions in water management. The use of climate normal is such an example and efforts are underway to link climate normal to hydrological normals calculated for rivers. Although such normal are not globally valid, they are regionally significant

We assume this comment refers to page 5 (not 4). We are aware that indices that require reference periods are regularly used for both scientific and management purposes. We investigated the use of

such indices in our preliminary assessments, but these indices had some issues and it proved impractical to develop a "one size fits all" solution. For example, large differences in temporal coverage prevented us from finding a reference period that is applicable around the world. These issues are outlined in the manuscript.

4,15 The notion that "time series indices cannot be extended when new data becomes available" is a serious constraint that limits the utility of the data set and the approach chosen. Moreover, a closed data set is aging fast. This needs to be discussed more in the paper. There is also the danger that such a data set becomes "orphaned" and continues to be used over a long period of time even when better data set have become available, similar like the RivDis Data of UNESCO more than 20 years ago that still is being used although it containes errors and is long-since superseded

We agree that there are many challenges related to maintaining the currency of a dataset. These challenges are various, and are both bureaucratic and technical. To this end, please see our response to the comments on Part 1 for a detailed discussion related to issues with orphaned data sets. Regarding the technical challenge of updating the dataset once new data are available, the index-computation for all indices in the manuscript does not depend on previous values and we consider this to be a relatively easy task. These points will be elaborated upon in the revised manuscript.

9,20 typ-o: ". . .to a year for which. . .."

Thanks for noting.

Quality control

The indices should be divided in those that actually express data quality and those that are significant for science applications such as water balance indications etc. A transparent approach should be taken to categorize data as a result of the data quality indices, making however clear that QC procedures cannot replace the responsibility of data owners and providers who in the first place provided data to data centres. The QC indices alone are already a valuable asset of the data set provided, guiding researchers and practitioners in the selection of data sets for their purposes.

The reviewer suggests providing an overall classification into indices into time-series suitable for specific applications based on the quality of the indices. Our approach has been to provide general information on quality rather than pre-determine their possible usage. That is, we (i) check the plausibility of daily values, (ii) provide the number of time-points that were used for computing index values at each time step (e.g. each year) and (iii) employ a set of previously suggested homogeneity tests, and for evaluating these results, we point to previously suggested criteria.

We consider that the "usefulness" of data is always very context dependent (which was also pointed out by referee # 2). For example, a study focussing on climate change detection will have different data requirements than a model validation exercise. Some studies will require highly selective quality criteria even though it results in few sites, while other studies will more readily trade-off quality for gauge density or record length. Others will implement or develop more rigorous quality procedures of their own by which to assess the data rather than be restricted by the basic checks we provide.

We appreciate that the above-mentioned quality control procedure may not be suitable for selecting data for all science applications and we will discuss this in the revised manuscript. We note also, that the scope of Earth System Sciences Data is on presenting data sets and not on developing new methodologies (which would be the case for a more sophisticated classification of data quality).

Discuss minimum length of time series needed for specific applications. For climate studies, i.e. WMO recommends a minimum length of 30 years.

We will mention this in the revised manuscript

The paper should discuss the robustness of the chosen indices on time series with differing qualities. A procedure could be proposed or recommended to check robustness of indices under varying conditions

Please see our response to the previous comments. We have implemented a basic set of quality checks on the indices and in the revised manuscript. We anticipate that there are numerous opportunities for proposing or developing new methods, but these are beyond the scope of the current paper. As a recommendation of the revised manuscript, we will identify the potential for further classification of robustness of indices and the need for methods to optimally classify the quality of time series in large streamflow data bases.

---

## Author Comment (AC2) · 26 Jan 2018

**Response to Reviewer #2**

We would like to thank Reviewer #2 for the fair and thoughtful comments which will help to improve the presented manuscript. In the following we provide point by point replies to the reviewer's comments. For the sake of clarity, we first repeat the reviewer's comments in blue before including our response.

General comments

I found this second, and (acknowledged) more difficult-to-do, part useful and publishable in general. But it reads less finished than the first part of the companion papers and my assessment is it still requires a bit more work on text and structure to be as influential in the end as perhaps can be on the community's use of the presented dataset. Again, of course the data quality control approach chosen, selection of indices for the dataset and the homogeneity tests chosen are subjective and pragmatic and not everyone needs to agree with it and it will never be possible to satisfy everyone's needs, so I don't see a point in criticising individual choices and decisions made here.

We would like to thank Reviewer #1 for this encouraging verdict of our approach.

What I think, however, would help the paper's acceptance and impact in particular regarding the use of the quality assessments given is to bring out the assessment of the result of these better, incl. for example a nicer clearer presentation and assessment of the summary statistics of quality flags, homogeneity tests and indices (Figs 4-7 are a bit uninspiring and very simple - maybe combining maps and box plots somehow or something like that) and more comment/assessment on the patterns on the maps that will allow at least some guidance to the data usage. Partly this is can perhaps be done by just a bit of a better organisation and presentation of the material that is there, but also by a more structured discussion, also linking to metadata from part 1.

We would like to thank Reviewer #2 for pointing out that the presentation of the quality control and homogeneity testing results could be improved. In the revised version the section concerned with the presentation of these results will be re-structured, aiming at a more in-depth presentation of the available information. To this end we will also consider creating more informative graphics (e.g. global time series of number of available stations, histograms/boxplots of time series lengths, regional summaries of data quality). In addition, we may include a more detailed discussion of how the information contained in the GSIM data can be used for data-selection. Nevertheless, we would like to emphasise that the scope of Earth System Sciences Data is the documentation of data and not their analysis.

It wasn't always clear to me, which information I will find in the dataset and which are only steps of production described here. This could be make clearer overall and in some cases removing repetitions may help (first an overview paragraph, then reading the same thing again later in the individual steps is unnecessary).

Thank you for noting that it was not always clear which information will be available in the data-stet. To mitigate this issue, we will consider including a new section "Description of data-files", in which we provide the necessary details.

Section 6 in particular is a bit random in what is covered and highlighted and thus presents not a strong conclusion. I suggest to give this some thought and better organise and bring out the highlights.

Thanks for emphasising the need to revisit the structure and the organisation of the text. We will consider this while revising the article. One topik that we will likely highlight, is the need for improved methods of quality checking of streamflow data. To this end we will revisit the approach used in this paper and point to avenues for future research in this field.

This is perhaps an editorial decision, but I find sections 3.2.1 to 3.2.15 not a correct nor a very useful text format as they contain neither a list nor paragraphed text and the use of the many subheadings is an unnecessary waste of space. Tables have been invented to reduce repetitive headings/descriptors. So why not a table with the name, abbreviation, unit, resolution and definition followed then by the more descriptive text paragraphs providing additional info. Alternatively, just a series of paragraphs always starting in similarly structured sentences would do as well.

We will likely reformat the content of theses sections into two tables as suggested by Referee #2. Any advise by the editor would be very welcome.

The selection of example studies is a bit random. Is it really necessary? This paper is supposed to describe the dataset created and it may be enough to use some of these references exemplary in a summary-motivation for the selection of indices or rather in a discussion on possible use of the dataset information.

In our evaluation, pointers to studies that make use of the considered indices do provide valuable background information to potential users. Nevertheless, we will consider moving this information into an appendix.

Generally, the manuscript will also require another careful proofread to correct several typos and some inconsistent formatting (italic or quotes for dataset flags/categories/...confusing!), some terminology (examples below), tenses (e.g. what 'was' done to the data - use part tense consistently - and what 'is' provided in the dataset - use present tense consistently) and notation (examples below - not an exhaustive list). In particular: see Journal's Manuscript guidelines for symbols, exponents and units (e.g change sec to s and make format exponents as superscripts (most figures))

The manuscript will be revised accordingly.

Selected specific comments

3.3.1. Isn't 'reference period' the more commonly used terminology (instead of base period)?

Depending on the body of literature considered, both terminologies are used. We will clarify this in the revised manuscript.

3.1.3 Requirements for number of valid data to estimate a 'reliable index'. These are very subjective, which I know is a necessary pragmatic solution. However, it creates a bias to less 'reliable' indices in climates streamflow gauging isn't possible or meaningful part of the year (seasonally dry climates and cold climates). This needs to be discussed.

The criteria mentioned in the paper are based on recommendations of ECA&D13. We acknowledge the caveat mentioned by Reviewer #2 and we will discuss this accordingly.

Harmonize the current mix of Q/C, Qc, qc, and define what is meant by it initially as common definitions vary.

Thanks for noting. We will revise accordingly.

27 typo: appraise

Thanks for noting.

Fig. 1 Typo in legend: 'equal', change axis label sec –> s and proper superscript (also in Fig. 3). I suggest to zoom in more as like this there is actually little to see.

The figure label will be revised. Note that the aim of this figure is to provide the "full view" on a time series. We will consider additional "zoomed" panel for the individual flags.

Fig. 3 Since a) daily data won't be provided by the dataset anyway if I understand correctly and b) one cannot see anything in the daily graph anyway, I suggest to remove it from Fig.3

The figure will be revised accordingly.

Similar to part 1, but perhaps even more so here, are the global maps. At that size and resolution it's impossible to see anything and not enough credit is given to why these difference may not simply reflect a lack of data but an inherent feature of the variable covered and which may not be present or measurable (see earlier comment). When I zoom in I see grids rather than station location points, but didn't read anything on gridding the point information. This is not acceptable and needs to be changed or very clearly described.

Please note that presenting >30000 locations on a world-map will always bee a compromise between providing the full spatial picture and the loss of some detail. If individual points should remain visible they have to have a certain size and will hence overlap in regions with high station density. We do, however acknowledge that the resolution of the figures is not optimal, and we will make sure to provide sufficiently resolved files for final production. In addition, we may opt for providing a set of regional figures in higher resolution in a supplement.